# Obesity Impairs Cognitive Function with No Effects on Anxiety-like Behaviour in Zebrafish

**DOI:** 10.3390/ijms241512316

**Published:** 2023-08-01

**Authors:** Alejandra Godino-Gimeno, Per-Ove Thörnqvist, Mauro Chivite, Jesús M. Míguez, Svante Winberg, José Miguel Cerdá-Reverter

**Affiliations:** 1Control of Food Intake Group, Department of Fish Physiology and Biotechnology, Instituto de Acuicultura de Torre de la Sal, IATS-CSIC, 12595 Castellon, Spain; a.godino@csic.es; 2Behavioural Neuroendocrinology, Department of Medical Cell Biology, Uppsala University, 751 23 Uppsala, Swedensvante.winberg@neuro.uu.se (S.W.); 3Behavioural Neuroendocrinology, Department of Neuroscience, Uppsala University, 751 24 Uppsala, Sweden; 4Centro de Investigación Mariña, Laboratorio de Fisioloxía Animal, Departamento de Bioloxía Funcional e Ciencias da Saúde, Facultade de Bioloxía, Universidade de Vigo, 36310 Vigo, Spain; mchivite@uvigo.es (M.C.);; 5Department of Anatomy, Physiology and Biochemistry, Swedish University of Agricultural Sciences, 750 07 Uppsala, Sweden

**Keywords:** obesity, overfeeding, fat, BMI, memory, anxiety, monoamines, zebrafish

## Abstract

Over the last decade, the zebrafish has emerged as an important model organism for behavioural studies and neurological disorders, as well as for the study of metabolic diseases. This makes zebrafish an alternative model for studying the effects of energy disruption and nutritional quality on a wide range of behavioural aspects. Here, we used the zebrafish model to study how obesity induced by overfeeding regulates emotional and cognitive processes. Two groups of fish (n = 24 per group) were fed at 2% (CTRL) and 8% (overfeeding-induced obesity, OIO) for 8 weeks and tested for anxiety-like behaviour using the novel tank diving test (NTDT). Fish were first tested using a short-term memory test (STM) and then trained for four days for a long-term memory test (LTM). At the end of the experiment, fish were euthanised for biometric sampling, total lipid content, and triglyceride analysis. In addition, brains (eight per treatment) were dissected for HPLC determination of monoamines. Overfeeding induced faster growth and obesity, as indicated by increased total lipid content. OIO had no effect on anxiety-like behaviour. Animals were then tested for cognitive function (learning and memory) using the aversive learning test in Zantiks AD units. Results show that both OIO and CTRL animals were able to associate the aversive stimulus with the conditioned stimulus (conditioned learning), but OIO impaired STM regardless of fish sex, revealing the effects of obesity on cognitive processes in zebrafish. Obese fish did not show a deficiency in monoaminergic transmission, as revealed by quantification of total brain levels of dopamine and serotonin and their metabolites. This provides a reliable protocol for assessing the effect of metabolic disease on cognitive and behavioural function, supporting zebrafish as a model for behavioural and cognitive neuroscience.

## 1. Introduction

There is currently an increasing volume of research focusing on the effects of metabolic disorders on cognitive and neurodegenerative processes [1]. Significantly, Alzheimer’s disease has been proposed as “type III diabetes” not only because of the high risk of dementia associated with type II diabetes but also due to the large amount of evidence on the existence of resistance to insulin in the brain during Alzheimer’s disease [2]. Diet and nutrition are also closely associated with mood disorders, including anxiety and depression, as well as other neuropsychiatric conditions [3]. Obesity, which is a global concern, challenges the health systems in Western societies and is also a major factor in metabolic syndrome, which has profound negative effects on brain structure [4] by impairing cognitive function and emotional states [1,5], thus exacerbating the development of learning/memory dysfunction and promoting anxiety-related responses [3,5].

In the last decade, zebrafish has emerged as a key model organism for behavioural studies [6] and neurological disorders [7,8], as well as for the study of metabolic diseases [9,10,11,12]. This makes zebrafish an interesting model for exploring the effects of energy disruption on behavioural aspects. Nutri-behavioural experiments mirroring those in mammalian species have also demonstrated the effect of diet-induced obesity (DIO), particularly high-fat and/or carbohydrate diets, on cognitive function [13] and anxiety-like behaviour [14]. There is some controversy about the behavioural effects of obesity according to the method involved in promoting obesity. Overfeeding with regular diets not only promotes obesity but also metabolically healthy fish, in contrast to high-fat-induced DIO, which generates hyperglycemia, ectopic lipid accumulation, and adipocyte hypertrophy, thus increasing visceral adipose tissue depots and inducing the expression of genes related to inflammation, fibrosis, and lipid metabolism [15]. However, overfeeding-induced obesity (OIO) with commercial food does induce deleterious effects on brain homeostasis and neuronal plasticity, thus promoting anxiety-like responses in zebrafish [16]. Nevertheless, similar experiments following an analogous approach to generating OIO in zebrafish were not able to replicate anxiety-like responses or find differences in aversive learning assays [17,18]. In the face of such controversy, the effects of OIO on anxiety-like behaviour and short- and long-term memory using aversive learning tests in zebrafish were explored. Moreover, the potential involvement of the central monoaminergic systems in obesity-induced behavioural and cognitive impairment was studied.

## 2. Results

### 2.1. Overfeeding Induces Obesity in Zebrafish

Overfeeding induced an increase in weight and length of OIO animals of both sexes (Figure 1). Significant differences in weight appeared earlier in females (2 weeks) than males (4 weeks), yet differences in length began as early as 2 weeks in both sexes (Figure 1A,B) and were maintained until the end of the experiment. After 8 weeks, at the beginning of the behavioural studies, OIO females and males were 62% and 27% heavier than control counterparts, respectively (Figure 1A). Significant differences in BMI were first observed after a month of overfeeding in females and males yet were only maintained at significant levels until the end of the experiment in OIO females (Figure 1C). Accordingly, OIO fish exhibited a higher lipid percentage (~200%) than control animals (Figure 2A) yet triglyceride levels were similar (Figure 2B). Together, biochemical and morphological data support overfeeding-induced growth and obesity in zebrafish (Figure 1D).

### 2.2. Obesity Does Not Promote Anxiety-like Behaviour in Zebrafish

In order to study anxiety-like behaviour in obese zebrafish, the geotaxis of OIO and control animals was assessed using the NTDT. Distance travelled was similar regardless of the feeding level in all tank areas (Figure 3A), yet the females covered less distance in the middle region of the tank (Figure 3A). The mean velocity was similar in every animal regardless of the feeding ration and sex (Figure 3B) yet remarkably, angular velocity was always depressed in OIO animals in all tank areas (Figure 3C) with no effects of sex (Appendix A). Time spent in each tank area was similar in every animal and arena (Figure 3D,E). Another two potential indicators of anxiety-like behaviour are the latency to the top zone and frequency of visits to this zone, but no difference was observed between feeding levels (Appendix A). Remarkably, sex had a significant effect on latency since males, regardless of the feeding level, took much less time to visit the upper region of the tank (Appendix A). Finally, effects on freezing bouts were not studied due to the limited number of fish displaying a single, brief freezing episode (two control female, one control male and 2 OIO males).

### 2.3. Obesity Does Not Alter Central Monoamine Levels in Zebrafish

To determine whether diet-induced obesity has any effect on brain serotoninergic and dopaminergic pathways, the central levels of the metabolites were analysed by HPLC. No significant differences in either monoamines and/or degradation metabolites induced by obesity were found (Figure 4).

### 2.4. Obesity Impairs Short-Term Memory in Zebrafish with No Effects on LTM

In order to investigate the effects of obesity on memory, two different tests for STM and LTM were carried out. Initial analysis indicated that both sexes responded similarly regardless of the feeding ratio (Appendix A), so both sexes were coupled in posterior analysis to increase the sample size. Regarding the STM, both groups exhibited similar PCS during the baseline phase (PCS = 0.5). Significant differences between the baseline and test phase demonstrated that both groups learned and memorized to avoid CS (Figure 5A). However, OIO fish exhibited significantly higher PCS levels than control fish, supporting less efficiency in short-term memory (Figure 5A). LTM showed that zebrafish consolidated information to avoid CS after 24 h; however, no significant differences were found between both feeding rates during the test phase (Figure 5B).

## 3. Discussion

In this study, the effect of OIO on anxiety-like behaviour and cognitive processes in zebrafish was explored. In the present study, the AB strain was used as it is commonly accepted to study obesity and obesity-related processes [15]. The increase in the feeding rate in OIO fish resulted in increased W, L, and BMI, in addition to elevated total fat percentage compared to control animals (~60%), supporting the obese condition of OIO fish, as previously reported [15,16,17,19]. The growth dynamic in the sexes appears to be different, with the result that females are more adept at increasing energy density when overfed than males, likely due to increased oocyte production. Similar results were obtained in our previous experiments [19,20] and by other authors [16]. In any case, the BMIs obtained in overfed fish were higher than those reported in previous experiments using similar overfeeding protocols [15,16]. Overall, the growth performance parameters, high BMIs, and much higher fat percentage levels support OIO in AB zebrafish.

The manner in which obesity is induced in zebrafish seems to have effects on behavioural responses and particularly anxiety-like responses. Zebrafish, either larvae or adult specimens, fed obesogenic diets containing high fat levels exhibit anxiety-like behaviours [14,21]; however, obese fish on regular diets at high ratios failed to exhibit such behaviour [17,18]. Due to the wide spectrum of behavioural responses displayed in NTDT by zebrafish [22,23,24], it was decided to enlarge the behavioural variables to corroborate previous results and study the activation of the central monoaminergic circuitry. Our results support the absence of anxiety-like behaviours in obese animals. Differences linked only to feeding levels were found in the angular velocity and mean turn angle in all three areas of the experimental arena. It is thought that these differences alone are not indicative of anxiety-like behaviours since they could be explained by the less agile movements of obese fish due to their rounded body shape in contrast to the lean control animals. However, such results reveal the sensitivity of geotaxis assays to decipher potential behavioural differences. In addition, sex-induced differences were also found in the total distance travelled and latency to the top zone of the arena by females which was inferior to that observed in males. Previous studies have demonstrated that females might exhibit lower locomotor activity regardless of body weight and length [25]. Once more, it is thought that the differences observed are not indicative of a sex-linked anxiety-like response.

Serotoninergic neurons in the posterior raphe of the caudal brain partially mediate anxiety-like behaviour in zebrafish [26], as well as sensory responsiveness during arousal [27]. The potential effect of OIO on the activation of central monoaminergic circuits was measured by quantifying the total 5-HT and dopamine levels and their primary metabolites in the whole zebrafish brain. The absence of significant differences supports previous results showing no effects of OIO on anxiety-like behaviour.

Obesity has been previously reported to impair cognitive function in vertebrates, including zebrafish [1,5,12]. Learning can lead to both STM and LTM [28], presumably managed and stored by different, yet interconnected, neuronal systems [29]. STM lasts from seconds to minutes and its formation depends on biochemical changes, whereas LTM lasts from hours to years and its formation relies on new protein synthesis [28,30]. In adult zebrafish, experiments focused on the study of social memory have estimated the period of 24 h as a long-lasting memory [31], although a similar social reward induced 36 h long-lasting memory in larval zebrafish [30]. Using the active avoidance test, the effect of OIO on learning capacity, STM, and LTM was studied. The results, expressed as the relation between the time expended in the CS versus total time (PCS), demonstrate that the animals were able to identify and avoid the CS. This implies that fish learn and memorize the association between the negative reward and the CS, as shown by a severe decrease in PCS after short periods. Animals were further able to retain this association for a 24 h period (LTM), although PCS values were higher than those in STM, showing a higher efficiency of STM when compared to LTM. OIO animals exhibited higher short-term PCS levels than control animals, suggesting that STM performance in obese animals is impaired. Although animals retained the stimulus/reward association for 24 h, corroborating the LTM, there were no differences in PCS values between both metabolic conditions, thus suggesting OIO had no effects on LTM.

The existence of STM and LTM has been demonstrated in zebrafish [31], and the effects of DIO on STM [13] and LTM have previously been studied [21]. DIO induced impairment of both STM and LTM, but OIO protocols had no effect on STM formation [17], although intergenerational effects of obesity have been reported [18]. Unfortunately, OIO effects on LTM response have not been studied. The present study focused on both STM and LTM, showing that OIO had significant effects on STM but no effects on LTM. Once more, the procedure involved in inducing obesity (DIO vs. OIO) in zebrafish appears to severely affect the cognitive process. Therefore, DIO exhibits consistent effects on both STM and LTM, whereas OIO had no effect on LTM and controversial effects on STM (Ref. [17] vs. the present study). Since diet composition appears to be critical for obesity-induced cognitive and behavioural impairment, the aforementioned controversy may stem from the dietary components involved in generating obesity by overfeeding. Previous studies involved solely freshly hatched artemia [32] or combined with commercially available fish food [17]. However, in this study, rotifer was used in combination with a dry diet which was specifically formulated for zebrafish husbandry, showing a markedly different nutritional composition to that used in previous studies [17,18,32]. Therefore, standardization of dietary protocols, diet composition, and ulterior studies on the effect of specific nutrients in obesity-induced cognitive and behavioural impairment appears to be critical for future investigations.

The underlying mechanisms responsible for the obesity-induced effects on cognitive performance are unknown; however, obesity is associated with decreased hippocampal neurogenesis [33], a function which has been shown to be enhanced during active learning [34]. Such obesity-induced effect on neurogenesis is probably mediated by neuro-inflammatory processes deriving from high-fat diets [33]. Interestingly, studies in DIO zebrafish have shown enhanced oxidative stress and decreased cell proliferation in the neurogenic niches of the diencephalon, and ventral and dorsomedial telencephalon [14,16], a fish homologue of the mammalian amygdala [34]. This complex pallial/subpallial structure plays a key role in Pavlovian fear conditioning [35]. The glutamatergic nuclei of the pallial amygdala are essential for the association of CS and US, which sends projections to the subpallial amygdala that serves as a primary output region projecting to different brain nuclei for the control of fear response [36,37,38]. Surgical or genetic ablation of the dorsomedial telencephalon promotes deficits in performing the avoidance response, thus indicating a role in the conditioned avoidance response [36,37]. LTM formation in zebrafish depends on protein synthesis and glutamatergic transmission since the incubation of larval zebrafish with protein synthesis inhibitors or NMDA receptor antagonists during training periods prevented or impaired memory formation, respectively [30]. Suggestively, the dorsomedial telencephalon in zebrafish has been shown to be rich in glutamatergic and GABAergic neurons, which mainly project to the subpallial and hypothalamic areas [39]. Since high-fat diets have been shown to induce changes in hippocampal glutamate metabolism and glutamatergic transmission [40], it is, therefore, conceivable that obesity-induced injuries in the glutamatergic population of the dorsomedial pallium may promote deficits in the aversive learning assays.

In summary, it can be observed that OIO has no effect on anxiety-like behaviour and LTM acquisition but impairs STM regardless of fish sex, revealing the obesity effects on cognitive processes in zebrafish. The results further suggest that these effects are entirely dependent on food protocols and/or diet composition. Obese fish exhibited no deficiency in monoaminergic transmission, revealed by the quantification of total brain levels of dopamine and 5-HT and their metabolites. Finally, a reliable protocol is provided to assess the effect of metabolic diseases on cognitive and behavioural function, thus supporting zebrafish as a model in behavioural and cognitive neuroscience.

## 4. Materials and Methods

### 4.1. Animals and Housing Conditions

Zebrafish (*Danio rerio*) AB strains, purchased from the zebrafish core facility at Karolinska Institutet, Solna, Sweden, were bred and housed in a recirculating Aquaneering system for zebrafish (Aquaneering, San Diego, CA 92126, USA) at Uppsala University Biomedical Centre. The system used copper-free non-chlorinated water from Uppsala municipality, 10% of which was exchanged daily. The temperature was set at 27 °C ± 1.5 SD, pH at 7.5–8, and the photoperiod was 14 h light and 10 h darkness, with lights on at 7 a.m. Fish were fed twice a day with rotifers and granulated feed (ZebrafishFeed, 63% protein, 13% fat, 1.8% fibre, and 1.2% ash, Sparos I&D, Olhao, Portugal) measured for each developmental stage. Subsequently, six-month-old zebrafish were distributed in 9 L tanks (~50:50 sex ratio). Body weight (W) and length (L) were measured, and body mass index was calculated (BMI = W(g)/L(cm)^2^) to ensure the absence of initial significant differences. Uppsala Animal Ethical Committee (permit 5.8.18-10125/2018) approved the use of animals in this study following the guidelines of the Swedish Legislation on Animal Experimentation (Animal Welfare Act SFS1998:56) and the European Union Directive on the Protection of Animals Used for Scientific Purposes (Directive 2010/63/EU).

### 4.2. Experimental Design

Two groups of 28 weight-graded fish were fed as previously mentioned at 2% (control; W = 0.504 g ± 0.975, L = 2.882 ± 0.178, and BMI = 0.060 ± 0.006) and 8% (overfeeding-induced obesity, OIO; W = 0.510 g ± 0.075, L = 2.854 ± 0.129, and BMI = 0.626 ± 0.007 SD) (Appendix A). No significant differences were detected at the beginning of the experiment as revealed by an unpaired *t*-test (*p* < 0.05). Biometrical samplings were carried out every two weeks for eight weeks (Figure 1A) as such intervals/times were reported to be sufficient for inducing overfeeding obesity in zebrafish [19,20]. The animals were tested for anxiety-like behaviour using a novel tank diving test (NTDT) (see below), and subsequently sampled for morphological parameters and individualised in tanks for acclimation for 24 h prior to memory tests (Figure 6A). Fish were first tested with short-term memory test (STM) and subsequently trained for four days for long-term memory tests (LTM) (Figure 6A). At the end of the experiment, fish were euthanised by immersion in benzocaine (5 mL/L) and biometrically sampled. Whole-body samples (n = 16; eight per treatment, sex ratio 50:50) were taken for total lipid content and triglyceride analysis. In addition, brains (n = 16; eight per treatment, sex ratio 50:50) were dissected for monoamine HPLC determination.

### 4.3. Novel Tank Diving Test (NTDT)

For the NTDT, fish from each experimental group (control, n = 24, and OIO, n = 24) were scooped from the home tank and placed in a trapezoidal tank (L: 27 cm × H: 15 cm × W: 5 cm, Aquaneering ZT180T) filled with 1.8 L of tap water as described in Figure 1B. Four fish were tested simultaneously on a shelf, but tank sides were shielded from light to avoid visual contact between subjects. An infrared light was placed behind the shelf to improve contrast during recordings. The NTDT took 10 min after 30 s of habituation. Video recording and tracking data acquisition were carried out with a Basler acA 1300 camera and Ethovision XT 16 (Noldus Inc., Wageningen, The Netherlands) (Figure 1B). All trials were performed on the same day (9 a.m.–12 a.m.). After the NTDT, each fish was individualised in a 2.8 L tank (Aquaneering ZT280T) for subsequent memory tests (see Section 2.4). In order to analyse the bottom-dwelling natural response to novelty, known as geotaxis [40], followed by a gradual exploration of the new environment, each arena was divided into 3 equal zones: top, middle, and bottom (Figure 6B); distance travelled, mean velocity, angular velocity, mean turn angle, and time spent moving were measured for each zone. Latency to the first top zone visit, frequency of visits, and the number of freezing bouts were also recorded.

### 4.4. Aversive Learning Methodology

To test the effect of obesity on cognitive traits, conditioned aversive learning was adopted. This behavioural paradigm involves associative learning to induce fish to link a visual cue to an aversive stimulus. Individual fish previously tested in the NTDT were randomly used for aversive learning experiments. Tests were carried out using four Zantiks AD units (Zantiks Ltd., Cambridge, UK), equipped with fully automated systems which require minimal intervention in order to avoid disturbances during trials. Each unit consists of a camera, computer, floor screen to display visual cues and a tank (H: 15 cm × W: 14 cm × D: 20 cm) equipped with two stainless steel plates connected to a power supply to deliver electric shocks. Zantiks AD technology allows the testing of 4 fish simultaneously, 2 control and 2 OIO (Figure 6C) [41]. Zantiks AD tanks were filled with 3 L of tap water at the same temperature as their home tank, and fish were then randomly placed individually in each compartment. In both short- and long-term memory tests, the visual stimulus consisted of two different patterns selected and named ‘check’ (black or white alternated squares) and ‘grey’ (solid light grey) (Figure 6C) [42,43,44]. The conditioned stimulus (CS) was ‘check’ while the non-conditioned stimulus (non-CS) was grey. The unconditioned stimulus (US) was a 9 V DC electric shock in both memory tests.

#### 4.4.1. Short-Term Memory Test (STM)

The STM testing protocol was an adaptation of the method employed by previous authors [41,43,44,45] and was tested in preliminary experiments. The script developed for the Zantiks web application to carry out this test is available on GitHub (https://github.com/Godino-Gimeno-A/aversive-learning/tree/Short-term-memory-test, accessed on 10 July 2023). As described in Figure 6D, the fish were acclimated for 10 min, displaying the visual stimuli on a half-divided background screen. The patterns were side-switched after 5 min. This habituation phase was followed by a baseline period, which is essential in order to ensure that fish avoid any place preferences. The baseline period lasted 30 min, during which the same switching protocol used in habituation was repeated 3 times. Conditioning was then conducted to induce the fish to associate visual stimuli with a negative reward. The CS (‘check’ or ‘grey’) was presented on the full screen for 1.5 s, paired with the US (9 V DC shock for 750 ms) and followed by 8.5 s of non-CS displaying (‘check’ or ‘grey’); this procedure was repeated 12 times. Finally, the subjects were tested to determine whether short-term memory was achieved, by observing whether the animals learned and remembered to avoid the CS. The test phase lasted 2 min, throughout which the fish were exposed to both the CS and non-CS, which switched position after 1 min in order to detect any place preference disturbances. The preferences for conditioned stimulus (PCSs) in the baseline and test phases were calculated as total time in CS/(time in CS + time in non-CS).

#### 4.4.2. Long-Term Memory Test (LTM)

The LTM in zebrafish using the Zantiks AD system was adapted for aversive learning based on the previous work on Drosophila [46]. The script for this test was developed after several preliminary trials and is available on GitHub (https://github.com/Godino-Gimeno-A/aversive-learning/tree/Long-term-memory-test, accessed on 10 July 2023). This protocol lasted 5 days (Figure 6E) and consisted of 4 training sessions followed by the test. The first training session was carried out the day after the STM test, and the previous CS was maintained for each fish. As shown in Figure 1, “conditioning”, the CS was displayed for 5 s, matched with the US (5 shocks of 9 V DC for 750 ms each) and repeated 12 times. Non-CS was then shown in the whole arena for a minute. After conditioning, an inter-trial interval (ITI), during which no stimulus was applied, was performed for 15 min to allow memory consolidation. Each training session comprised 6 consecutive trials (conditioning + ITI). Training trials were carried out at 9 a.m. for 4 consecutive days. Twenty-four hours after the last training session, the test was performed for 2 min following the same protocol as that used in STM tests, and subsequently, the PCSs were calculated.

### 4.5. Total Lipids and Triglycerides

Whole-body samples from both groups (8 of each genotype, sex ratio 50:50) were lyophilised, weighed, and homogenised into fine particles. Lipid extraction was carried out following Folch’s method [47] and was quantified gravimetrically. Subsequently, the percentage of total lipids was calculated for each sample. A fraction of each sample (~50 mg) was used in the triglyceride colourimetric assay kit from Cayman Chemical (Cayman Chemical Company, Ann Arbor, MI, USA) and the percentage of triglycerides was subsequently calculated.

### 4.6. Monoamine Analysis

At the end of the study, OIO and control brains were dissected (8 of each genotype, sex ratio 50:50), and stored at −80 °C. The samples were thawed in ice-cold phosphate buffer (pH = 7.4, 1 M) and homogenised using an ultrasonic disruptor. Levels of 5-hydroxytryptophan (5HTP), serotonin (5-hydroxytryptamine, 5-HT), and 5-hydroxy indoleacetic acid (5HIAA) were determined using HPLC with electrochemical detection [48]. Additionally, metabolites of the dopaminergic pathway were measured: L-dihydroxyphenylalanine (L-dopa), dopamine (DA), and dihydroxyphenylacetic acid (DOPAC). The mobile phase (pH = 2.95) content was 63.9 mM NaH2PO4, 0.1 mM Na2EDTA, 0.80 mM sodium 1-octanesulfonate, and methanol 15.3% (*v*/*v*). HPLC equipment consisted of a Jasco pump (PU-2080); a reversed-phase analytical column (Phenomenex, Kinetex C18, 5 µm, 100 Å, 150 mm × 4.6 mm); an M5011 double analytical cell (first electrode: +40 mV; second electrode: +340 mV) to oxide analytes and an ESA Coulochem II detector. Results were obtained using ChromNAV 1.12 software (Jasco Corporation, Tokyo, Japan). Levels of each metabolite per total protein content in the brain (measured with BCA), 5HIAA/5HT ratio, and DOPAC/DA ratio were calculated.

### 4.7. Statistics

Statistical analyses were performed with GraphPad Prism 8. A two-way ANOVA and Tukey’s multiple comparison tests were performed for morphological parameters. Regarding the NTDT, significant differences between the control and OIO groups and between sexes were investigated with two-way ANOVA. No multiple comparison tests were run since the interactions were not significant. In the STM and LTM tests, significant differences between the PCS baseline and the tests within both experimental groups and between control and OIO were assessed by mixed-effects two-way ANOVA and Sidak’s multiple comparisons tests. An unpaired *t*-test was employed in studying differences between groups in the percentage of total lipid content and triglycerides, and in monoamines levels. All data are represented as mean ± SEM.

## Figures and Tables

**Figure 1 ijms-24-12316-f001:**
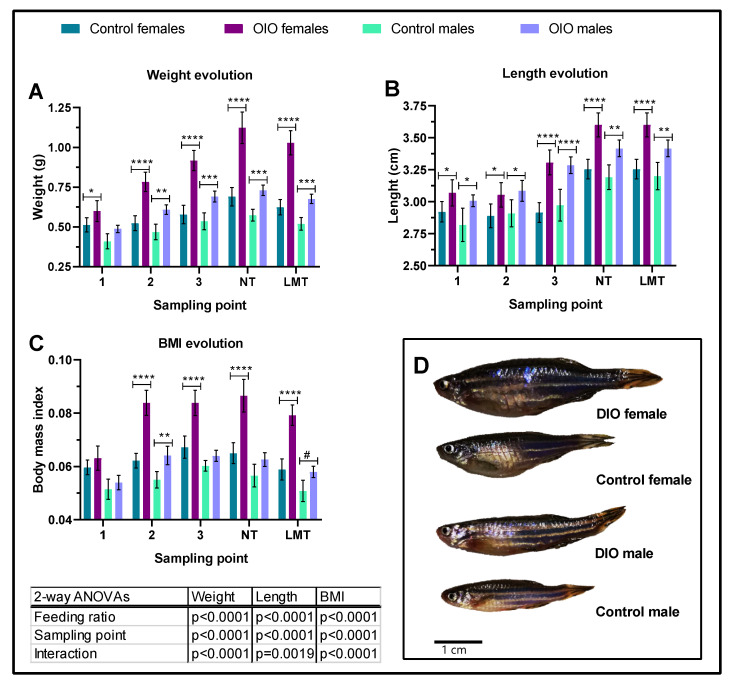
Experimental dynamics of morphological parameters. Fish were sampled every two weeks in 1, 2, 3, and NT (NTDT testing day) and one week between NT and LTM (long-term memory test probe phase). (**A**) Weight, (**B**) length, and (**C**) biomass index (BMI). (**D**) Representative images of control and OIO females and males at the end of the study. Data are expressed by mean ± SEM and analysed by two-way ANOVA and Tukey’s multiple comparison test. Asterisks indicate significant differences between OIO and control within the same gender (* *p* ≤ 0.05, ** *p* ≤ 0.01, *** *p* ≤ 0.001, **** *p* ≤ 0.0001), and # *p* = 0.051.

**Figure 2 ijms-24-12316-f002:**
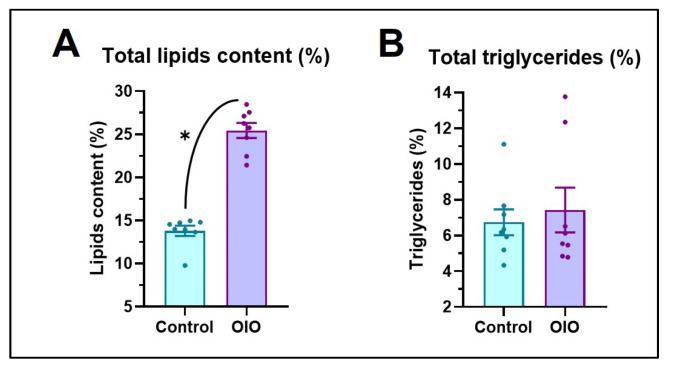
(**A**) The percentage of total lipids. (**B**) Triglyceride content (n = 8 per treatment, sex ratio 50:50). Data are expressed by mean ± SEM and analysed by an unpaired *t*-test (* *p* ≤ 0.05).

**Figure 3 ijms-24-12316-f003:**
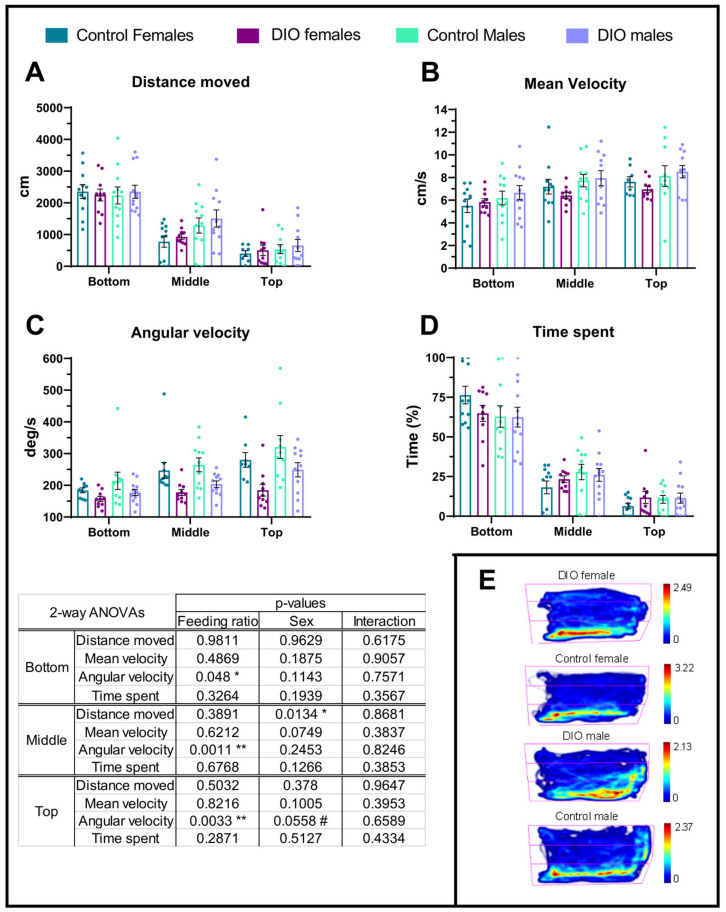
Novel tank diving test (NTDT). Anxiety-like behaviours were measured in three zones: bottom, middle, and top. (**A**) Distance travelled, (**B**) mean velocity per zone, (**C**) angular velocity, and (**D**) time spent in each zone. (**E**) Representative heatmaps of fish movements. The colour scale represents the cumulative time spent in each of the previously defined zones: bottom, middle, and top. Data are represented by mean ± SEM and analysed by two-way ANOVA. Asterisks indicate significant differences (* *p* ≤ 0.05, ** *p* ≤ 0.01), and the hashtag indicates an apparent significant difference (# *p* = 0.051–0.06).

**Figure 4 ijms-24-12316-f004:**
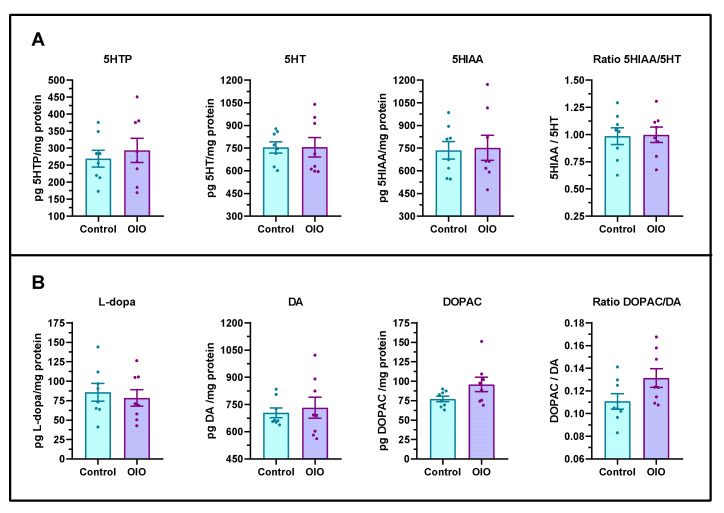
Brain serotoninergic and dopaminergic metabolites. (**A**) 5-hydroxytryptophan (5HTP), serotonin (5HT), and 5HIAA/5HT ratio. (**B**) Levels of L-dihydroxyphenylalanine (L-dopa), dopamine (DA), and dihydroxyphenylacetic acid (DOPAC) and DOPAC/DA ratio (n = 8 each per treatment, sex ratio 50:50). Results are expressed in pg of metabolite per mg of total protein. Data are represented by mean ± SEM analysed by an unpaired *t*-test.

**Figure 5 ijms-24-12316-f005:**
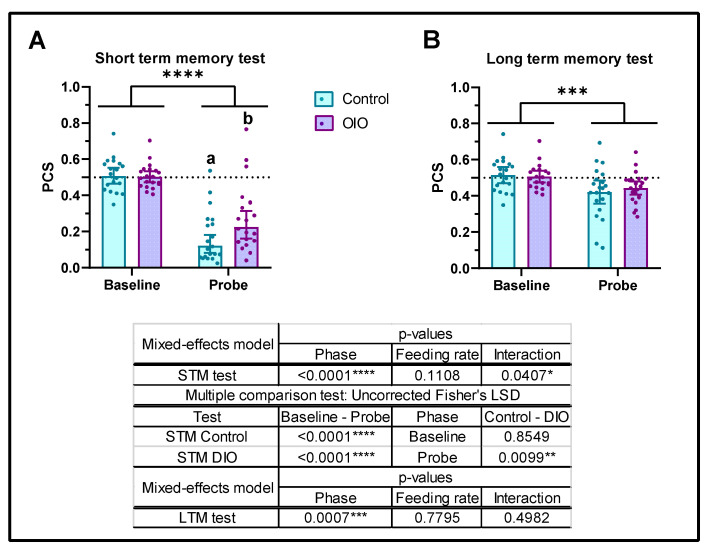
Effects of obesity on short- and long-term memory. (**A**) Short-term memory (STM) and (**B**) long-term memory (LTM). PCS, preference for conditioned stimulus. Data are represented by mean ± SEM. The dotted line indicates no stimulus preference (PCS = 0.5); low and high PCS mean less or more time spent in conditioned stimulus (CS), respectively. Results were analysed by repeated measures mixed-effect model and, when interaction was significant, uncorrected Fisher’s LSD test for multiple comparisons. Asterisks indicate significant differences between phases, baselines, and probes (* *p* ≤ 0.05, ** *p* ≤ 0.01, *** *p* ≤ 0.001, **** *p* ≤ 0.0001), and letters indicate significant differences between control and OIO probes.

**Figure 6 ijms-24-12316-f006:**
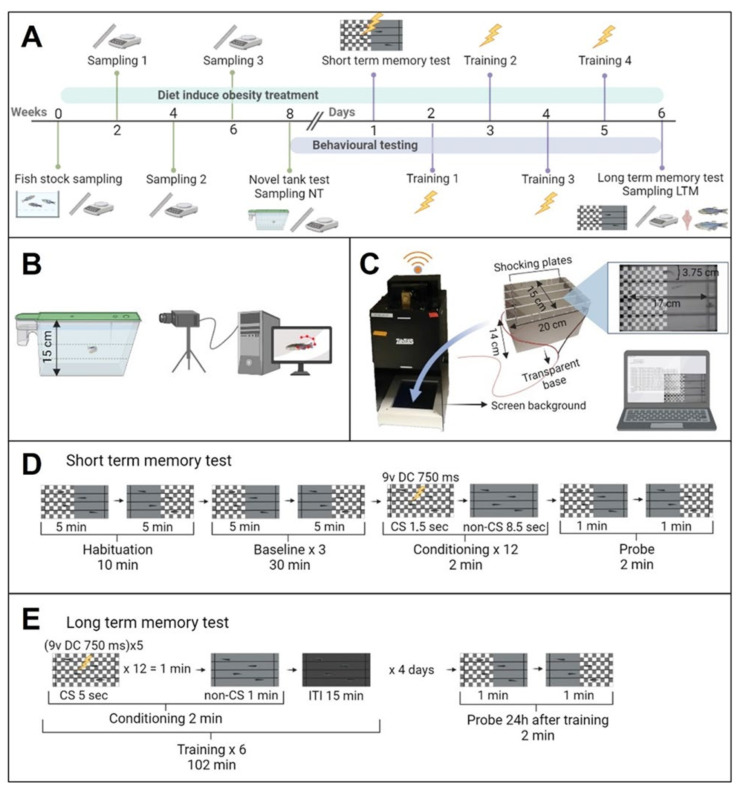
(**A**) Experimental design diagram. Fish from the stock were weight-graded and divided into control and overfeeding-induced obesity (OIO) groups. Animals were fed at 2 and 8%, respectively, for 8 weeks and sampled for morphological parameters every two weeks. Following the feeding protocol, the behavioural tests began with a novel tank diving test (NTDT). Animals were subsequently isolated in individual tanks for 24 h (n = 16 for each treatment). Fish were then tested for short-term memory (STM), and subsequently trained for 4 days for LTM tests. Finally, the fish were euthanised and sampled for morphological parameters. Eight fish were used for lipid quantification and the brains of the remaining eight fish were dissected for monoamine determination by HPLC. (**B**) Novel tank diving test setup. Fish were placed in trapezoidal tanks, and their trajectories were acquired for 10 min using Ethovision 16 XT. (**C**) Zantiks AD unit representation, used for aversive learning tests. (**D**) Short-term memory test protocol. Fish were habituated for 10 min to the experimental conditions in which the screen was half-divided into ‘check’ and ‘grey’ switching positions every 5 min. The potential preference for any background was recorded for 30 min (baseline). The same schedule used for the habituation period was then repeated 3 times. Throughout the conditioning phase, the conditioned stimulus (CS) was displayed for 1.5 s and paired to the unconditioned stimulus (US, 9 V DC shock for 750 ms) followed by 8.5 s of non-conditioned stimulus (‘check’ or ‘grey’). This protocol was repeated 12 times and learning and STM were tested for 2 min. (**E**) Long-term memory test protocol. Fish were initially trained for 4 consecutive days. The conditioning phase was followed by an inter-trial interval (ITI) of 15 min to allow memory consolidation, and the probe phase was performed 24 h after the last training session. The CS was displayed for 5 s and linked to the US (5 shocks of 9 V DC, 750 ms each). This protocol was repeated 12 times and subsequently non-CS was shown for 1 min followed by the ITI (absence of stimulus). Conditioning + ITI were repeated 6 times for each training session. Finally, 24 h after the last training session, the probe phase took place. In both STM and LTM tests, the preference for conditioned stimulus (PCS) was calculated as total time in CS/(time in CS + time in non-CS). Significant differences between baseline and probe PCSs were analysed. The significantly lower probe PCS, the memory enhancer, was achieved.

## Data Availability

https://data.mendeley.com/datasets/bdm7pzfh45/1 (accessed on 10 July 2023).

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
