# Peer review of "Obesity Impairs Cognitive Function with No Effects on Anxiety-like Behaviour in Zebrafish"

_ijms, 2023, doi:10.3390/ijms241512316_

Round 1
Reviewer 1 Report
The authors used in this study the zebrafish model in order to determine how obesity induced by overfeeding regulates emotional and cognitive processes. Two groups of fish were used for this study, the first one was fed at 2% the second one and 8% (overfeeding-induced obesity, OIO) for 8 weeks and tested for anxiety-like behaviour using the novel tank diving test (NTDT). The abstract is clear, introduction part is explanatory. Results and Discussion chapters are well written. Experimental part is well explained. Authors used modern methods in order to obtain accurate results.
A reliable protocol is provided to assess the effect of metabolic diseases on cognitive and behavioural function.
There is a difference between the number of fish in abstract of the paper and sub-chapter 4.3. Novel tank diving test -please correct. So the title of this paper” Obesity impairs cognitive function without effects on anxiety-like behavior in zebrafish” is in concordance with obtained results.
Author Response
Dear reviewer 1
We are very grateful for your comments on the paper. As indicated, we have corrected the number of experimental fish in the abstract.
Reviewer 2 Report
IJMS 2526924
Manuscript #2526924: Obesity impairs cognitive function with no effects on anxiety-2 like behaviour in zebrafish
Authors: Alejandra Godino-Gimeno, Per-Ove Thörnqvist, Mauro Chivite, Jesús M. Míguez, Svante Winberg and José Miguel Cerdá-Reverter
The authors feed 6-month-old zebrafish females and males a high-fat diet and try to establish a causal relationship between obesity and behavioral and cognitive disorders and also try to associate with possible changes in 5HTP, serotonin and 5HIAA levels and metabolites of the dopaminergic pathway. The data obtained do not confirm this association, and only show a lower locomotor activity associated with obesity.
Specific Comment:
The four paragraphs of the discussion are very long and it is difficult to distinguish the interpretation of the results obtained with the large amount of information from the literature used.
Too much information confuses the understanding of the data.
The authors could divide all the paragraphs, rewriting the text focusing on the essential information and highlighting the negative data obtained.
Author Response
Dear Reviewer 2:
We are very grateful for your suggestions on the paper that have significantly improved the quality of the MS. In our experiments, obesity was induced exclusively by overfeeding the same diet with no differences in fat composition. Results demonstrate that obese animals did not display anxiety-like behaviour. Locomotor differences were only associated with feeding levels in the angular velocity and mean turning angle, although we thought that these differences per se were not indicative of anxiety-like behaviours as they could be explained by the less agile movements of obese fish. On the contrary, obese animals were less efficient in short-memory tests liking the negative reward with the conditioned stimulus (CS). We are aware of the extension of the provided information in the discussion. However, we believe that it is necessary to provide an understandable and coherent framework for the discussion of the results thus integrating and fitting our results with those already published. We agree that paragraphs need to be shortened to make the section easier to read and have shortened them, accordingly.
Round 2
Reviewer 2 Report
The authors have separated four paragraphs from the discussion which are very long, yet a great deal of information from the literature remains in the text. The text could be rewritten.
And it would be necessary to introduce a Conclusion at the end of the text
Author Response
To whom it may concern.
We appreciate your kind comments on the paper. According to the template provided by the journal, the conclusions are written in Section 5, immediately after the Materials and Methods. We agree with you, and we think, honestly, that the position of this section 5 is not the right one and it should be immediately after the Discussion. As this section 5 is not mandatory, we have moved the paragraph summarising the conclusions to the end of the Discussion.
As required we have rewritten the discussion eliminating some potential redundant information in the paper to reduce complexity and make this section easier to read.
We have also re-numbered the references according to the new content of the paper